# Wearable 12-Lead ECG Acquisition Using a Novel Deep Learning Approach from Frank or EASI Leads with Clinical Validation

**DOI:** 10.3390/bioengineering11030293

**Published:** 2024-03-21

**Authors:** Fan Fu, Dacheng Zhong, Jiamin Liu, Tianxiang Xu, Qin Shen, Wei Wang, Songsheng Zhu, Jianqing Li

**Affiliations:** 1School of Biomedical Engineering and Informatics, Nanjing Medical University, Nanjing 211166, China; 2The First Affiliated Hospital of Nanjing Medical University, Nanjing 210029, China; echo_yaya77@163.com; 3The Jiangsu Engineering Research Center of Province Intelligent Wearable Monitoring and Rehabilitation Device, Nanjing Medical University, Nanjing 211166, China; 4The Engineering Research Center of Intelligent Theranostics Technology and Instruments, Ministry of Education, Nanjing Medical University, Nanjing 211166, China; 5The State Key Laboratory of Bioelectronics, School of Instrument Science and Engineering, Southeast University, Nanjing 210096, China

**Keywords:** deep neural network, EASI lead system, electrocardiogram, 12-lead ECG reconstruction

## Abstract

The 12-lead electrocardiogram (ECG) is crucial in assessing patient decisions. However, portable ECG devices capable of acquiring a complete 12-lead ECG are scarce. For the first time, a deep learning-based method is proposed to reconstruct the 12-lead ECG from Frank leads (V_X_, V_Y_, and V_Z_) or EASI leads (V_ES_, V_AS_, and V_AI_). The innovative ECG reconstruction network called M2Eformer is composed of a 2D-ECGblock and a ProbDecoder module. The 2D-ECGblock module adaptively segments EASI leads into multi-periods based on frequency energy, transforming the 1D time series into a 2D tensor representing within-cycle and between-cycle variations. The ProbDecoder module aims to extract Probsparse self-attention and achieve one-step output for the target leads. Experimental results from comparing recorded and reconstructed 12-lead ECG using Frank leads indicate that M2Eformer outperforms traditional ECG reconstruction methods on a public database. In this study, a self-constructed database (10 healthy individuals + 15 patients) was utilized for the clinical diagnostic validation of ECG reconstructed from EASI leads. Subsequently, both the ECG reconstructed using EASI and the recorded 12-lead ECG were subjected to a double-blind diagnostic experiment conducted by three cardiologists. The overall diagnostic consensus among three cardiology experts, reaching a rate of 96%, indicates the significant utility of EASI-reconstructed 12-lead ECG in facilitating the diagnosis of cardiac conditions.

## 1. Introduction 

Heart disease is the leading cause of mortality worldwide [1]. Electrocardiogram (ECG) monitoring serves as an effective means for the early detection of cardiovascular disease [2]. In clinical practice, the 12-lead ECG plays a pivotal role in assessing and guiding patient management decisions [3]. In order to record prolonged cardiac activity, ambulatory ECG was introduced in 1961 [4]. However, due to its influence on daily life, which stems from the number and placement of recording points and its relatively short recording duration (most 20 to 48 h [5]), there is an urgent need for new measurement methods to capture long-term cardiac activity.

The wearable ECG, while meeting long-term monitoring and comfort demands [6], falls short of meeting clinical requirements as the standard 12-lead ECG. Existing wearable ECG devices predominantly capture single leads (two electrodes [7] or optical sensors [8]) or three-lead ECG (four electrodes [2] or five electrodes [9]). Compared with standard 12-lead ECG, wearable ECG offers limited intuitive cardiac information (as shown in Figure 1) and is currently primarily used to diagnose arrhythmias [10,11,12]. To our knowledge, no specific diagnostic standards have yet been established for wearable ECG in clinical practice. The reconstruction of a standard 12-lead ECG from wearable ECG data can enhance the clinical utility of wearable ECG. As a result, the reconstruction of 12-lead ECG from a reduced number of leads has become a research hotspot.

The theoretical foundation for the reconstruction of ECG was established by Frank [13,14,15] and Dower et al. [9,16,17,18]. Frank et al. introduced the Frank-XYZ orthogonal spatial vector ECG (V_X_, V_Y_, and V_Z_) [13], but it is not suitable for dynamic cardiac monitoring. Based on Frank et al.’s theory, Dower et al. proposed the EASI lead system (V_ES_, V_AS_, and V_AI_), which is suitable for dynamic acquisition, and theoretically demonstrated the feasibility of reconstructing the 12-lead ECG using EASI leads [16]. As shown in Figure 1, the EASI system consists primarily of four electrodes (E-A-S-I) that can capture three bipolar leads (V_ES_, V_AS_, and V_AI_), each containing information from both the transverse and coronal planes. Notably, not all three-lead systems can reconstruct a 12-lead ECG. For example, limb leads only contain information from the coronal plane and do not provide the necessary information to derive chest leads, theoretically lacking the feasibility to derive precordial leads [17,19,20]. Dower et al. introduced the “Dower universal transform” method, which achieves a linear transformation of EASI data to derive 12-lead ECG using a biased matrix [16]. Field et al. then enhanced the “Dower universal transform” coefficients originally proposed by Dower et al. [18], and Nelwan et al. observed significant differences between 12-lead ECGs reconstructed using improved EASI coefficients and the recorded ones [21]. Similarly, Schreck et al. employed a straightforward nonlinear approach to construct a universal matrix to reconstruct missing leads [22,23]. The foundational leads used to reconstruct the remaining 12 ECG leads initially included I, aVF, and V2 [22], with subsequent work incorporating I, II, and V2 [23]. This method represents an ideal “one-size-fits-all” solution but may not adapt well to interferences from factors such as equipment, biological variations, and environmental conditions [24].

The least squares regression (LSR) method was used by Trobec et al. to estimate the transformation to generate a 12-lead ECG from three differential leads (DLs) [25]. This method yielded the best results in generating 12 leads from the three DLs proposed by the authors, with an average correlation coefficient of 0.954. However, this method exhibited a lower correlation coefficient of 0.71 in lead aVL, and the root mean square error reached 115.3 μV in lead V5. Their study aligns with the approach of Dower et al., resulting in limited generalization capabilities. Mulyadi et al. proposed reconstructing the 12-lead ECG using a segment-based approach (divided into P, QRS, and T segments) through LSR [26]. Unfortunately, abnormal ECG can exhibit phenomena such as P wave disappearance, QRS-wave distortion, and low-amplitude T wave, which can cause reconstruction failure. Despite attempts to use neural networks to synthesize ECG [27], including the application of focused time-delay neural networks used for speech recognition to ECG reconstruction [28], as reported in their results, the generalizability of ECG reconstruction has improved but still requires further enhancement.

The EASI-lead ECG represents a simplified expression of cardiac status, while the 12-lead ECG provides a richer and more clinically informative representation. This result is analogous to the task of machine translation, where understanding the semantics of one language and translating it into another is required. The Transformer model and its variants are currently among the state-of-the-art models in the field of machine translation. Furthermore, they have also shown good performance in time-series forecasting [29,30,31,32]. With the assistance of attention mechanisms, they can uncover hidden pairwise temporal dependencies between time points. Zhou et al. introduced the application of the Transformer model to the prediction of long sequence time series, using its attention mechanism to capture long-term dependencies within the sequence [29]. However, it is challenging for attention mechanisms to directly identify reliable dependencies from scattered time points [31].

In this study, we analyze ECG signals from a multicycle perspective. Sinus ECG exhibits quasiperiodic behavior. However, the conduction of abnormal cardiac electrical activity is influenced by the current cardiac cycle and the increased excitability of ectopic rhythm points or the reentrant excitement from the last cycle, presenting a multicycle pattern. Consequently, the detected abnormal ECG signal results from the superimposition of sinus rhythms and ectopic rhythms, exhibiting multicycle characteristics. However, raw ECG sequences have a one-dimensional structure that captures changes only between adjacent time points, making it challenging to explicitly extract both types of variation simultaneously. We employ Fourier Transformation to dissect 1D time series into several segments based on the ECG frequency composition, stacking them into a 2D structure. At this juncture, the rhythms within each segment predominantly represent within-cycle variations, whereas the variances in the ECG at identical positions across segments are shaped by between-cycle variations. This enables us to represent within-cycle and between-cycle variations concurrently in a 2D space, resulting in temporal 2D variations.

Motivated by the abovementioned considerations, we propose a multichannel-based 2D-variation ECG reconstruction network (M2Eformer). This network comprises two primary modules as follows: the 2D-ECGblock and the ProbDecoder. With the support of the 2D-ECGblock, M2Eformer can identify the multicyclic nature of ECG sequences and fuse information into the attention-based ProbDecoder to achieve target leads. We evaluated the algorithm’s performance in publicly available databases [33] using quantitative metrics such as the Pearson coefficient r (*Pr*) and mean absolute error (*MAE*), as well as macro-level evaluations provided by cardiologist annotations. Furthermore, regarding practical application value, we collected synchronous EASI and 12-lead ECG from cardiac patients who required 12-lead ECG monitoring. We analyzed the consistency in the diagnoses made by cardiac experts for the reconstructed and recorded 12-lead ECG.

As illustrated in Figure 1, this document establishes a mapping relationship between EASI and the 12-lead ECG using the deep learning model M2Eformer.

For the first time, a deep learning-based ECG reconstruction network is presented, which deeply extracts latent cardiac information from EASI leads and reconstructs a standard 12-lead ECG consistent with the diagnostic practices of cardiac experts. This provides a feasible approach to the application of wearable ECG for clinical diagnosis.We propose a 2D-ECGblock module for the reconstruction network that transforms time-domain signals into multiperiod 2D tensors based on spectral energy. This module simultaneously extracts dependent information from both within-cycle and between-cycle components in the ECG. Additionally, we designed the ProbDecoder module, which employs a sparse attention mechanism to achieve ECG reconstruction in a residual-like manner.We conducted a clinical diagnostic validation study of 25 cases using a 12-lead ECG reconstructed from EASI leads. Next, focusing on four cardiac conditions, namely, atrial fibrillation, atrial flutter, coronary artery disease, and myocardial infarction, which require 12-lead ECG monitoring, three experts were invited to participate in a double-blind diagnostic experiment comparing the reconstructed 12-lead ECG with standard recorded ones. The overall consistency coefficient reached 96%.

The remaining parts of the paper are structured as follows: Section 2 outlines the framework of this paper, encompassing the composition of the dataset, the network architecture, and the evaluation methodologies employed. Section 3 presents the results. Then, Section 4 provides the discussion. Finally, Section 5 summarizes the conclusion.

## 2. Materials and Methods

The general framework of this study, as depicted in Figure 2, comprises three modules as follows: data preparation, model construction, and results analysis. The aim is to reconstruct a standard 12-lead ECG using EASI leads (V_ES_, V_AS_, and V_AI_). The following two databases were used in this research: a publicly available database (Frank-XYZ + 12-lead ECG) [33] and a self-constructed database (EASI leads + 12-lead ECG), each serving different experimental purposes including the algorithm comparison experiment (Task 1) and the EASI practicality analysis experiment (Task 2).

To construct the M2Eformer model, we initially calculated the correlation coefficient distribution between input signals (V_X_, V_Y_, and V_Z_, or V_ES_, V_AS_, and V_AI_) and target signals on the training set. The lead with the highest correlation was selected as the input for the corresponding ProbDecoder model. Subsequently, we used M2Eformer to reconstruct the 12-lead ECG. Finally, a results analysis was conducted. The details of each component are further elaborated below.

### 2.1. Databases

In this study, we used the publicly available PhysioBank Physikalisch-Technische Bundensanstalt Diagnostic (PTB-DN) ECG database [33] to compare the performance of the Task 1 algorithm. The main reasons for this choice are as follows: 1. the PTB-DN database includes synchronous Frank-XYZ leads and standard 12-lead ECG, with Frank-XYZ leads forming the theoretical basis for EASI; 2. the PTB-DN database is the largest publicly available database known to contain both synchronous Frank-XYZ leads and standard 12-lead ECG, comprising 549 records from 290 subjects; and 3. many previous studies on ECG reconstruction have also utilized this database [16,22,23,27,28], which facilitates our algorithm comparison experiments.

PTB-DN data were sampled at a rate of 1000 Hz with a 16-bit resolution, and the least significant bit represented 0.5 μV. Before use, all ECG records were preprocessed in 8 s windows, involving a 50 Hz notch filter and 20th-order polynomial filtering to eliminate powerline noise and baseline drift. In order to eliminate the influence of high-frequency noise, local regression smoothing filtering was applied with a smoothing window of 10 sample points. Furthermore, despite preprocessing, some records still contained significant artifacts (ECG drowned by noise or existing severe wandering baseline) or missing information (missing leads or diagnostic information) and were excluded from this study. The data composition used for the algorithm evaluation is detailed in Table 1.

Among these, each category of ECG records was roughly divided into training, validation, and test sets in a ratio of approximately 3:1:1 [29,34]. It should be noted that the data for the training and test sets were strictly derived from different individuals.

In order to validate the reliability of the 12-lead ECG reconstructed by EASI for monitoring purposes, Task 2 involved the collection of synchronized EASI leads (V_ES_, V_AS_, and V_AI_) and standard 12-lead ECG. As depicted in Figure 1, thirteen electrodes were attached to the patient’s body, where ten electrodes were used to capture the 12-lead ECG, and four electrodes, with one overlapping electrode A and V6, were utilized for capturing EASI leads (V_ES_ = V_E_ − V_S_, V_AS_ = V_A_ − V_S_, and V_AI_ = V_A_ − V_I_). Heart disease patients were arranged for ECG collection at the First Affiliated Hospital of Nanjing Medical University. As shown in Table 2, the types of heart diseases among the patients included atrial flutter (1 case), atrial tachycardia (2 cases), myocardial infarction (3 cases), and coronary heart disease (9 cases). There were also ten healthy participants from Nanjing Medical University. Data were collected using medical equipment (NaLong RAGE-18P) with a sampling rate of 1000 Hz. Healthy individuals were monitored for 10 min, while patients were monitored for 5 min. It is important to emphasize that the selected types of heart disease required joint assessment using a 12-lead ECG. Ethics approval was obtained from the Nanjing Medical University Ethics Committee.

### 2.2. Multichannel 2D-Variation ECG Reconstruction Network (M2Eformer)

Figure 3 illustrates the network architecture of the proposed 12-lead ECG reconstruction model. M2Eformer consists of two modules, namely, the 2D-ECGblock and the ProbDecoder module. In the 2D-ECGblock module, the ECG data are adaptively transformed into a 2D representation based on frequency domain energy, thus enabling simultaneous extraction of within-cycle and between-cycle variations. In the ProbDecoder module, initial sparse-attention calculations are performed on the input signal (Max Correlation Lead) to extract relevant information from the ECG. Subsequently, in the Encoder–Decoder Attention layer, the extracted data are fused, providing the foundational knowledge for the reconstruction of target leads.

As shown in Figure 3, the input to M2Eformer consists of a three-lead ECG represented by V_X_, V_Y_, and V_Z_. For a cardiac sequence of length *L*, the original 1D structure is denoted as *X*_1D_ ∈ R*^L^*
^× 3^. The collected ECG vectors represent the projection of the vectorcardiography at that moment onto the coordinate axes of the electrodes and the cardiac dipole. Therefore, to extract cardiac information at time *t*, we designed the multichannel fusion layer, and the computational method is as follows:(1)$$X_{1D}^{dmodel}={\mathrm{Conv}1d}_{3\times 3}\left(X_{1D}^{\mathrm{Ein}}\right)$$

By mapping the original three-channel ECG into a high-dimensional vector $X_{1D}^{dmodel}$ and simulating the distribution of vectorcardiograms at time *t*, we enhanced the model’s generalization capability.

In order to capture between-cycle variations in the ECG sequences, it is essential to first identify their periods. Inspired by the work of Hu et al. [34], we designed the adaptive 2D unfolding module, referred to as the 2D-ECGblock. This method utilizes the Fast Fourier Transform to identify the highest *m* frequency bands with the highest energy in the ECG sequence, as shown below:(2)$$A_{f_{*}}=\mathrm{Amp}\left(\mathrm{FFT}\left(X_{1D}^{dmodel}\right)\right),\;f_{*}\in \left\{1,\cdots ,L/2\right\}$$
(3)$$\left\{f_{1},\cdots ,f_{m}\right\}=\mathrm{Top}\left(\mathrm{Avg}\left(A_{f_{*}}\right)\right)$$
(4)$$p_{i}=\frac{T}{f_{i}},\;i\in \left\{1,\cdots ,m\right\}$$

In the above context, FFT(·) represents the Fast Fourier Transform, and Amp(·) is used for the calculation of the amplitude. Af∗ denotes the amplitudes calculated for each frequency band, and their mean across the *dmodel* dimensions is obtained through the Avg(·) function. Given the sparsity in the frequency domain, we sought to avoid the noise impact of irrelevant high frequencies; thus, we selected only the top *m* amplitudes, obtaining the most significant frequency bands {*f*_1_, …, *f_m_*} along with their corresponding amplitudes {Af1,⋯,Afm}. These selected frequency bands correspond to the durations of *k* period lengths {*p*_1_, …, *p_m_*}. Due to the conjugate symmetry in the frequency domain, we only use frequencies within the {1, …, *L*/2} range. Based on the selected period lengths {*p*_1_, …, *p_m_*} and frequencies {*f*_1_, …, *f_m_*}, we can reconstruct the 1D time sequence X1D∈ℝT×dmodel into a 2D tensor using the following formula:(5)X2Di=Reshape2D, pi, fi(Padding(X1Ddmodel)), i∈{1,⋯,m}

In the above formula, Padding(·) extends the time sequence by padding zeros along the time dimension to evenly divide X1Ddmodel into *f_i_* segments along the time dimension. Next, *p_i_* and *f_i_* represent the number of rows and columns in the resulting 2D tensor after transformation, where each row represents between-cycle variation and each column represents within-cycle variation. X2Di∈ℝpi×fi×dmodel is the *i*-th 2D tensor obtained based on frequency *f_i_*. After transformation, an efficient Inception block was applied [35] to process the 2D tensor, denoted as Inception(·). In our implementation of Inception(·), we include 2D convolution kernels of three scales including 1, 3, and 5. The calculation formula is as follows:(6)X2Di^=Inception(X2Di), i∈{1,⋯,m}

The Inception(·) module here is shared among m layers of $X_{2D}^{i}$ tensors to improve parameter efficiency.

Finally, we need to transform the $\left\{\hat{X_{2D}^{1}},\cdots ,\hat{X_{2D}^{m}}\right\}$ back into 1D representations for the next layer and perform information fusion. Inspired by Wu et al. [31], the amplitude of each frequency band reflects its relative importance. Here, we base the fusion on the transformed m 1D tensors after amplitude-based fusion. The formula is as follows:(7)$$\hat{X_{1D}^{i}}={\mathrm{Reshape}}_{1D,\;p_{i},\;f_{i}}\left(\hat{X_{2D}^{i}}\right),\;i\in \left\{1,\cdots ,m\right\}$$
(8)$$X_{1D}^{\mathrm{Eout}}=\sum_{i=1}^{m}\hat{A_{f_{i}}}\times \hat{X_{2D}^{i}},\;\hat{A_{f_{*}}}=\mathrm{Softmax}\left(A_{f_{1}},\cdots ,A_{f_{m}}\right)$$

Due to the within-cycle and between-cycle dependency information encapsulated in the *m* highly structured 2D tensors, the 2D-ECGblock can extract multiscale temporal 2D variations through the Inception module. Compared with the original Transformer, which obtains interelement dependencies through attention mechanisms, the 2D-ECGblock enables more efficient representation learning.

The ProbDecoder has two input components. The first part of the input consists of one of the Frank-XYZ leads (V_X_, V_Y_, and V_Z_). In the training dataset, we computed the *Pr* between the Frank-XYZ leads and the target lead, as shown in Figure 4. When training the corresponding model, the lead from V_X_, V_Y_, or V_Z_ with the highest correlation coefficient to the target lead is selected as the input for the ProbDecoder. According to the statistical results in Figure 4, the final correspondence for the ProbDecoder input is as follows: I-X, II-Y, III-Y, aVR-X, aVL-X, aVF-Y, V1-Z, V2-Z, V3-Z, V4-Z, V5-X, and V6-X. From the graph, it can be observed that the leads with the highest correlation are negatively correlated with the standard 12 leads, specifically aVR, V1, V2, V3, and V4. This is because aVR-X, V1-Z, V2-Z, V3-Z, and V4-Z represent vectors located on the opposite side of the heart with opposite polarities.

First, we encode the input $X_{1D}^{\mathrm{Din}}$ for the ProbDecoder:(9)$$Q,K,V=\mathrm{Linear}\left(\mathrm{Conv}1d\left(X_{1D}^{\mathrm{Din}}\right)\right)$$
where *Q*, *K*, and *V,* respectively, represent the query, key, and value matrices in the Transformer, with *K* being the same size as Q (*L_K_* = *L_Q_* = *L*). Since the input ECG for the ProbDecoder itself is sparse, with a small portion of physiologically significant cardiac signals and a larger portion of baseline signals, we were inspired by Zhou et al. [29] to propose a Probsparse self-attention calculation for the encoded cardiac data. *Q* only needs to perform dot products with ln(*L_K_*) key matrices randomly, and the remaining *L_K_*–ln(*L_K_*) pairs are filled with zeros. The calculation process is as follows:(10)$$\left[\begin{array}{c} x_{1}^{1}\\ \vdots \\ x_{L}^{1} \end{array}\begin{array}{ccc} 0 & \cdots & 0\\ \vdots & \ddots & \vdots \\ 0 & \cdots & 0 \end{array}\begin{array}{c} x_{1}^{L}\\ \vdots \\ x_{L}^{L} \end{array}\right]=\mathrm{Padding}\left(Q\times K_{\ln L_{K}}\right)$$
(11)$$Q=\left[\begin{array}{ccc} q_{1}^{1} & \cdots & q_{1}^{dmodel}\\ \vdots & \ddots & \vdots \\ q_{L_{Q}}^{1} & \cdots & q_{L_{Q}}^{dmodel} \end{array}\right],K=\left[\begin{array}{ccc} k_{1}^{1} & \cdots & k_{1}^{dmodel}\\ \vdots & \ddots & \vdots \\ k_{\ln L_{K}}^{1} & \cdots & k_{\ln L_{K}}^{dmodel} \end{array}\right]$$

In the computed *L* × *L* matrix, only ln(*L*) columns have numerical values. Therefore, in the ProbDecoder, self-attention only needs to calculate O(*L* × ln(*L*)) dot products. Max-mean measurements are performed on the computed *L* × *L* matrices:(12)$$M_{h}=\max\left\{x_{h}^{1},\cdots ,x_{h}^{L}\right\}-\frac{1}{L}\sum_{j=1}^{L}x_{h}^{j},\;h\in \left\{1,\cdots ,L\right\}$$

Next, based on the sorting of {*M*_1_, ⋯, *M_L_*}, we select the top-*u* $\left\{x_{u}^{1},\cdots ,x_{u}^{dmodel}\right\}$ vectors of *Q* to form $\bar{Q}$, where *u* = *C* × ln *L*.

Here, *C* is a hyperparameter, and it was chosen as *C* = 5 based on results from [29]. The self-attention matrix computed for the sparse matrix $\bar{Q}$, *K*, and *V*, is also sparse, with the remaining rows of the *V* matrix filled with the mean of that row. This approach helps to emphasize the importance of the positions, where the ECG waveforms are located while reducing the model’s focus on baseline waveforms. The final Probsparse self-attention matrix still has a size of *L* × *L*, which is calculated as follows:(13)$$Attention=\left\{\mathrm{Softmax}\left(\frac{\bar{Q}K^{T}}{\sqrt{dmodel}}\right)V,\mathrm{Mean}\left(V\right)\right\}$$

As mentioned above, periodic variations are extracted from the V_X_, V_Y_, and V_Z_ three-lead ECG signals through the 2D-ECGblock. Based on this information, we perform attention mechanism calculations in the Encoder–Decoder layer and correct central ECG waveforms in the value matrix, achieving reconstruction of the target lead electrocardiogram in a residual-like manner. Therefore, based on this attention calculation, a new value matrix $\bar{V}$ is computed as follows:(14)$$\hat{V}=\mathrm{Norm}\left(Attention+X_{1D}^{\mathrm{Din}}\right)$$
(15)$$\hat{Q},\hat{K}=\mathrm{Linear}\left(\mathrm{Conv}1d\left(X_{1D}^{\mathrm{Eout}}\right)\right)$$

The second part of ProbDecoder’s input is the output $X_{1D}^{\mathrm{Eout}}$ from the 2D-ECGblock. After a linear transformation, new query $\bar{Q}$ and key $\bar{K}$ matrices are obtained. The ECG waveform correction is performed in the Encoder–Decoder Attention layer, and after passing through a feedforward layer and a linear layer, the target lead ECG is obtained as follows:(16)$$\mathrm{Target}=\mathrm{Linear}\left(\mathrm{Feed}\left(\mathrm{Softmax}\left(\frac{\hat{Q}{\hat{K}}^{T}}{\sqrt{dmodel}}\right)\hat{V}\right)\right)$$

In order to ensure that the training process of each lead ECG reconstruction network does not interfere with each other, we trained 12 separate M2Eformer models, each dedicated to reconstructing the corresponding lead ECG signal.

### 2.3. Evaluation

Based on previous research, we used the Pearson coefficient r (*Pr*) and the mean value of absolute error value (*MAE*) to quantify the differences between the predicted ECG leads and the recorded leads. *Pr* measures the degree of linear correlation between two sets of data, variables *X* and *Y*, and is calculated as follows:(17)$$Pr=\frac{\sum_{i=1}^{n}\left(X_{i}-\bar{X}\right)\left(Y_{i}-\bar{Y}\right)}{\sqrt{\sum_{i=1}^{n}{\left(X_{i}-\bar{X}\right)}^{2}}\sqrt{\sum_{i=1}^{n}{\left(Y_{i}-\bar{Y}\right)}^{2}}}$$
where *X* represents the reconstructed ECG leads, *Y* represents the recorded ECG leads, and *n* denotes the duration of each record.

*MAE* provides a better reflection of the actual amplitude error in the reconstructed ECG, with smaller values indicating greater reconstruction precision. It is calculated as follows:(18)$$MAE=\frac{1}{n}\sum_{i=1}^{n}\left|X_{i}-Y_{i}\right|$$

## 3. Results

### 3.1. A Comparison of Training Results

The proposed M2Eformer model employs an attention-based Transformer architecture. It utilizes a single layer of the 2D-ECGblock module as the Encoder and a single layer of the ProbDecoder module as the Decoder. The embedding dimension was set to 512. We initially conducted grid search experiments on the PTB-DN validation set with epochs of 100 and 200, and learning rates of 0.001, 0.0001, and 0.00001. An epoch of 100 and a learning rate of 0.0001 were selected, taking into account both training speed and reconstruction performance. Subsequent experiments were conducted on the validation set to evaluate various configurations, including the number of layers in the 2D-ECGblock module (0, 1, and 2 layers) and the ProbDecoder module (1, 2, and 4 layers), as well as different embedding dimensions (64, 256, and 512). Based on these tests, we ultimately selected a setup with one layer for the 2D-ECGblock module, one layer for the ProbDecoder module, and an embedding dimension of 512. The batch size for training was set to 200, determined by the GPU’s memory capacity of 24 G. To prevent model overfitting and enhance the generalization capability of the training model, we employed the Dropout function as the regularization method, with the dropout rate set to 0.1. The training process utilized the Adam optimizer and the MSE (Mean Squared Error) Loss function [30,31,32,34].

The loss curves on the validation set are depicted in Figure 5, where gray represents the original Transformer, blue represents the T-Transformer, which embeds the 2D-ECGblock into the Transformer while keeping the Decoder unchanged, and red represents the proposed M2Eformer.

From Figure 5, we can observe that in leads aVR, aVL, V1, and V4, the proposed M2Eformer achieves a lower validation loss in the validation set, significantly outperforming both the Transformer and T-Transformer. With other leads, the convergence results are relatively close. The minimum validation loss values and their corresponding best epochs for each of the three models are listed in Table 3. As indicated in Table 3, the proposed M2Eformer has a slightly higher validation loss on lead V5 compared with the Transformer (0.0001) but achieves better or consistent results on the remaining leads. Moreover, Figure 5 demonstrates that M2Eformer does not exhibit a noticeable overfitting phenomenon in all leads despite its slower convergence compared with the other two frameworks.

In Figure 5, when comparing the validation loss between the Transformer and the T-Transformer, we notice that the Transformer exhibits a more pronounced overfitting issue (especially in leads aVR, V2, and V4). The cause of overfitting may be attributed to the attention mechanism in the encoder failing to capture reliable temporal dependencies within the signal [31]. Our parameter analysis revealed that the total number of parameters in the Transformer (4.2 million) is smaller than that in the T-Transformer (13.4 million). This result suggests that the phenomenon of overfitting is not caused by excessively large model parameters, further confirming the effectiveness of the 2D-ECGblock in extracting hidden cardiac information.

### 3.2. ECG Reconstruction Effect Comparison

To validate the performance of M2Eformer, we compared it with various algorithms using two key metrics including *Pr* and *MAE*. The algorithms compared included Transformer, T-Transformer, as well as algorithms mentioned in previous studies, such as Linear transformation [16,22,23] and least squares regression (LSR) [25]. We also included the commonly used Long Short-Term Memory (LSTM) network for comparison in time series tasks [36,37,38].

The results demonstrate that several methods used in the experiments can reconstruct the 12-lead ECG, with superior overall performance achieved by deep learning-based approaches. Table 4 and Table 5 present the *Pr* and *MAE* between the reconstructed ECG and the recorded ECG obtained using these six algorithms in the test dataset, where the ratio of training, validation, and testing was set at 3:1:1.

Table 4 reveals that the proposed M2Eformer exhibits the best overall reconstruction performance for the 12-lead ECG (total *Pr* = 0.8785), followed by the T-Transformer (total *Pr* = 0.8579). M2Eformer surpasses the Transformer in performance across leads II-V1 and V3-V6 for each lead, underscoring the efficacy of the 2D-ECGblock.

Although LSR achieves the highest *Pr* in a few leads (aVR, V3, V5, V6), the average correlation coefficient in leads III, aVL, and aVF is less than 0.7. This finding indicates that the ECG reconstructed by LSR in leads III, aVL, and aVF deviates significantly from the recorded ECG (as shown in Figure 6), especially in lead III, where the amplitude difference in the S wave reaches 1mV. This divergence could potentially lead to a misdiagnosis (e.g., patients with reduced ECG amplitudes suggestive of myocardial injury in cases of coronary artery disease).

The *MAE* can characterize the actual errors in the predicted values, with smaller values indicating a smaller amplitude difference between the reconstructed ECG and the recorded ECG. In Table 5, M2Eformer only exhibits the lowest *MAE* in a few leads (aVL, aVF, V4). However, its overall *MAE* to reconstruct the 12-lead ECG is the lowest (total *MAE* = 0.0511). Although the Linear method achieves the lowest *MAE* in leads II, V1-V3, its performance in terms of *Pr* in Table 4 is not outstanding. This is because the Linear method better fits the waveforms with larger amplitudes (Q and S waves) in these four leads. As depicted in Figure 7, the amplitudes of the R and S waves reconstructed in leads III and aVL for Linear assessment differed significantly. This result can also result in a misdiagnosis by cardiologists (e.g., diagnosing coronary artery disease as myocardial injury).

In general, considering the results in Table 4 and Table 5, M2Eformer achieves the best overall performance in reconstructing the 12-lead ECG, and the T-Transformer shows improvement compared with the original Transformer. This result demonstrates that the 2D-ECGblock meets our expectations for effectively extracting ECG information.

Figure 8 presents box plots of *Pr* in the test set for the Transformer, the T-Transformer, and M2Eformer. The mean *Pr* for each lead in Table 4 is also represented in the figure as squares (□). In Figure 8, M2Eformer shows a more concentrated *Pr* distribution in most leads (II, III, aVR, aVL, and V3-V6), with higher mean and median values. This result indicates that the ECG reconstructed by M2Eformer shows more consistent waveform changes (synchronously rising and falling) with the recorded ECG.

The Transformer performs better in lead I, but compared with the other two algorithms, it does not show statistically significant differences at a confidence level of *p* = 0.05. The Transformer only shows statistically significant superiority (higher mean) in leads aVF and V2. By comparing the Transformer and M2Eformer training processes (Figure 5 and Table 3), we observe that M2Eformer achieves a lower loss in the validation set and does not exhibit overfitting. We believe that this might be due to the limited size of the validation dataset, which may not fully reflect the real training process.

In summary, Figure 8 further demonstrates the superior performance of M2Eformer.

The performance of M2Eformer in reconstructing the ECG is shown in Figure 9. This segment (1.25 s) of the ECG data was collected from a patient with a myocardial infarction. The red line represents the reconstructed ECG, while the black line represents the recorded ECG. We presented a 10 s segment containing this ECG snippet to a cardiac specialist for diagnosis. The diagnosis based on the reconstructed ECG (red line) indicates “old anterior myocardial infarction (V1-3 leads exhibit QS morphology)” and “lateral myocardial ischemia (ST-segment depression in leads I, V5-6)”. The diagnosis based on the recorded ECG (black line) is “anterior myocardial injury (poor R-wave progression in V1-3 leads)” and “lateral myocardial ischemia (ST-segment depression in leads I, V5-6)”. Among them, “old anterior myocardial infarction” and “anterior myocardial injury” correspond to the same cardiac injury, but the expression is different. In the reconstructed ECG, there is a noticeable discrepancy in the 0.5–1 s region of leads V2 and V3 compared with the recorded ECG. However, these differences are mainly in terms of amplitude, with their waveforms being nearly synchronous, indicating that M2Eformer captures the periodic variations in the ECG signal and reflects them in the output. However, there is room for improvement in M2Eformer in terms of the extraction and representation of waveform amplitude information.

In the test dataset, we performed an analysis of the consistency between the diagnostic results of the reconstructed ECG and the recorded ECG, as shown in Table 6. We selected 10 s ECG segments that demonstrated the highest average Pr between the reconstructed and recorded 12-lead ECG on that record, resulting in 61 segments of reconstructed ECG and 61 segments of recorded ECG. Diagnostics were performed using a double-blind method. When the diagnostic results of the recorded ECG and the reconstructed ECG for the same segment were consistent with the cardiologists, we considered the reconstructed ECG to have no impact on clinical diagnosis. Overall agreement (OvA) is a method used to assess the consistency in diagnoses among three experts. For example, in the case of the reconstructed and recorded ECG segments, if two experts arrive at the same diagnostic conclusion, the segment is considered to have consistent OvA, even if the third expert’s diagnosis diverges.

As shown in Table 6, the percentage of consistency for cardiologist 1 was 93.4%, for cardiologist 2 was 93.4%, and for cardiologist 3 was 90.2%. We calculated the overall consistency among the three experts, which reached 96.7%, with only two cases of inconsistency among the diagnoses of healthy individuals; the reasons for these inconsistencies are examined in the Section 4.

### 3.3. EASI Leads to 12-Lead ECG

In order to further validate the reliability of the 12-lead ECG reconstruction through the EASI lead configuration for monitoring purposes, we conducted simultaneous data collection of EASI leads and standard 12-lead ECG. Furthermore, to comply with clinical requirements, we selected patients with various cardiac conditions that require a combined 12-lead diagnosis. Ultimately, we obtained effective ECG data from 10 healthy individuals and 15 patients, including those with atrial fibrillation, atrial flutter, coronary artery disease, and myocardial infarction. Moreover, due to the limited sample size, we employed 5-fold cross-validation for our analysis. The hyperparameters of the M2Eformer model (epoch = 100, learning rate = 0.00001, batch size = 200) remained unchanged.

The experimental results are presented in Figure 10, where (a) shows the histogram distribution of Pr between the reconstructed ECG and recorded ECG, (b) displays the boxplot distribution of *Pr* and *MAE* between the reconstructed ECG and recorded ECG, and (c) illustrates the consistency results of the annotations by cardiac experts for the reconstructed ECG and recorded ECG.

Figure 10a reveals that in more than half of the leads (lead I, aVR, V2, V4, V5, and V6), the proportion of *Pr* greater than 0.8 exceeds 90%. Among the remaining leads, in leads II, aVF, and V3, more than 80% of the *Pr* values are greater than 0.8, while in leads III, aVL, and V1, the proportion of *Pr* values exceeding 0.8 is around 70%. Combining Figure 10a,b, we observe that in more than half of the leads (lead I, II, aVR, aVF, and V2–V6), the median Pr exceeds 0.9, with even lead V1 having a median *Pr* of 0.9044. Although Figure 10b shows that the median *Pr* values for leads III and aVL are below 0.9, their median *MAE* values are 0.0326 and 0.0302 mV, indicating small differences in amplitude.

We engaged three cardiologists to annotate the reconstructed ECG and the recorded ECG for a macro evaluation. We selected 10 s ECG segments for each record, comprising 25 segments of reconstructed ECG and 25 segments of recorded ECG. The diagnostic results are shown in Figure 10c, with individual diagnosis consistency rates of 96%, 96%, and 92% for the three experts. Notably, among them, inconsistent samples from cardiologist 2 and cardiologist 3 are interlaced. Importantly, the samples identified as In-CS by cardiologists 1 and 2 are identical, indicating that the OvA classification for this particular sample is marked as In-CS. Conversely, the samples identified as In-CS by cardiologist 3 do not overlap with those deemed In-CS by cardiologists 1 and 2, thereby not impacting the final OvA analysis. Therefore, the OvA among the diagnostic results from three cardiac experts achieved 96% (24/25).

In Figure 10c, for the only sample with In-CS OvA outcomes, the reconstructed ECG interpretations varied as follows: healthy individuals with variants or old anterior interwall myocardial infarction (cardiologist 1), coronary heart disease (cardiologist 2), and healthy (cardiologist 3). Conversely, the recorded ECG was unanimously classified as healthy by all three cardiologists. This ECG was obtained from a patient who had returned to sinus rhythm following ablation for atrial flutter. Therefore, in this study, there is a certain discrepancy between the reconstructed and recorded 12-lead ECG. Nonetheless, the high consistency observed in the one-versus-all (OvA) outcomes (96%) underscores the substantial adjunctive value of EASI-reconstructed 12-lead ECGs in the clinical diagnosis of atrial fibrillation, atrial flutter, and coronary artery disease.

## 4. Discussion

This study is the first of its kind to propose a deep learning-based ECG reconstruction network that reconstructs 12-lead ECG from EASI leads, enabling EASI leads to help diagnose a wider range of cardiac diseases. In this study, the designed novel ECG reconstruction network involves the following key components: 1. the 2D-ECGblock, which simultaneously extracts within-cycle and between-cycle dependencies from input ECG, and 2. the ProbDecoder, which is a carefully designed generation component using Probsparse self-attention mechanisms to achieve residual-like ECG reconstruction. Furthermore, we conducted clinical diagnostic validation of the reconstructed 12-lead ECG on our self-established database. The diagnostic results of the cardiologists indicate that the EASI-reconstructed 12-lead ECG has the potential to assist in the diagnosis of atrial flutter, atrial fibrillation, coronary artery disease, and myocardial infarction. Conversely, the use of EASI leads in isolation offers minimal assistance in the diagnosis of these four conditions.

Linear regression (Linear) [15,16,17,18] and least square regression (LSR) [15,25,26] are commonly used methods for the reconstruction of 12-lead ECG. Attention-based deep learning networks have achieved promising results in time-series prediction tasks [29,30,31,32]. Consequently, this study presents M2Eformer, a novel attention-based model for 12-lead ECG reconstruction, and conducts a comprehensive performance comparison with traditional methods, including Linear and LSR, widely utilized in prior research. The results in Table 4 and Table 5 and Figure 6 and Figure 7 demonstrate that the proposed M2Eformer outperforms LSR and linear methods in the overall performance of Frank-XYZ reconstruction of 12 leads on the PTB-DN database. In Table 4, M2Eformer performs best in the reconstruction of the ECG for most leads, although its performance is slightly lower than that of LSR in the reconstruction of the aVR, V3, V5, and V6 leads. In future studies, the complexity of the parameters required for each lead’s reconstruction can be explored. Additionally, the dataset can be further expanded to optimize the M2Eformer model and enhance the reconstruction performance of each lead model.

In Table 6, there were differing opinions among the three cardiologists regarding the annotations for two healthy individuals. For the first inconsistent sample, the cardiologists provided different annotations for the reconstructed ECG including bundle branch block (cardiologist 1) and healthy (cardiologists 2 and 3). However, their annotations for the recorded ECG were myocardial infarction (cardiologist 1), incomplete right bundle branch block (cardiologist 2), and healthy (cardiologist 3). According to the PTB-DN database records, this ECG was collected from a healthy individual. The inconsistency in the diagnostic results for this sample is primarily attributed to cardiologist 2’s interpretation of the recorded ECG as RBBB, due to a significant error in the diagnosis by cardiologist 1 for this sample. In the second inconsistent sample, the cardiologists provided different annotations for the reconstructed ECG including possible high lateral myocardial ischemic injury (cardiologist 1), myocardial ischemia (cardiologist 2), and possible myocardial injury (cardiologist 3). However, their annotations for the recorded ECG were possible myocardial ischemic injury (cardiologist 1) and healthy (cardiologists 2 and 3). According to the database records of the PTB-DN, this ECG was also obtained from a healthy individual. Based on the comprehensive annotations of the three experts, the reconstructed ECG was annotated as “myocardial injury” (indicating myocardial infarction), while the recorded ECG was annotated as “healthy”. The diagnostic inconsistencies observed in this sample may stem from the data imbalance within the PTB-DN database, characterized by a discrepancy between myocardial infarction cases (213) and healthy individuals (57). This imbalance, favoring myocardial infarction instances, might have led to the inadvertent integration of myocardial infarction-related features into the reconstructed ECG, culminating in erroneous annotations.

The primary limitation of the proposed model is that the loss function employs a generic calculation method and does not adjust specifically for abnormal ECG waveforms, such as incorporating the error between the R waves of the reconstructed and recorded ECG into the training loss for backpropagation. Another limitation of this study is the relatively small sample size of our self-constructed database, and the issue of data balance needs further resolution. In our subsequent work, we will address this issue by collecting a more diverse range of clinical data.

## 5. Conclusions

This paper explores the clinical diagnostic value of using 12-lead reconstructed ECG through EASI leads for wearable ECG monitoring. A novel network architecture designed for ECG reconstruction, called M2Eformer, is proposed. This model utilizes the 2D-ECGblock to synchronously extract information regarding within-cycle and between-cycle dependencies. Information fusion is achieved through a specially designed ProbDecoder, enabling the reconstruction of a 12-lead ECG. The experimental results demonstrate that M2Eformer achieves the best overall reconstruction performance for 12 leads (*Pr* = 0.8785 and *MAE* = 0.0511), with *Pr* values higher than traditional methods such as the LSR and Linear methods (0.0517 and 0.0593, respectively). Expert annotations obtained from the recorded data (overall consistency of 96%) suggest the potential value of the reconstructed 12-lead ECG in aiding the clinical diagnosis of conditions such as atrial flutter, atrial fibrillation, coronary artery disease, and myocardial infarction.

## Figures and Tables

**Figure 1 bioengineering-11-00293-f001:**
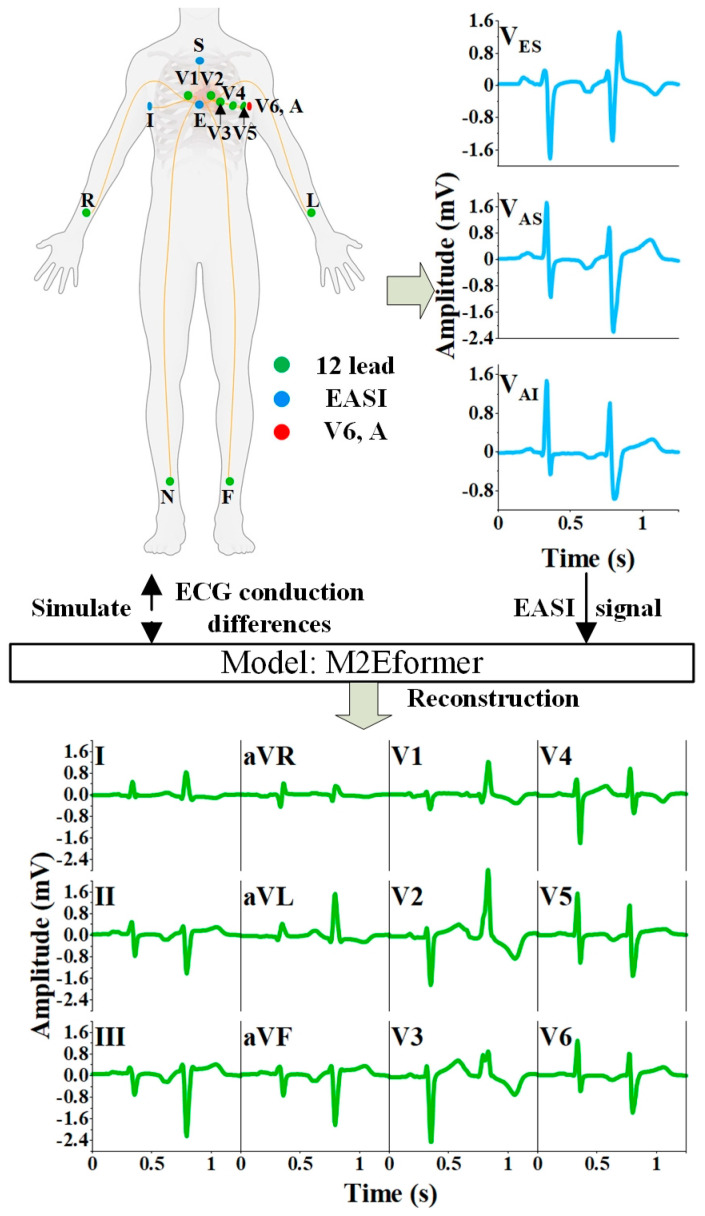
Schematic illustration of the 12-lead ECG reconstruction.

**Figure 2 bioengineering-11-00293-f002:**
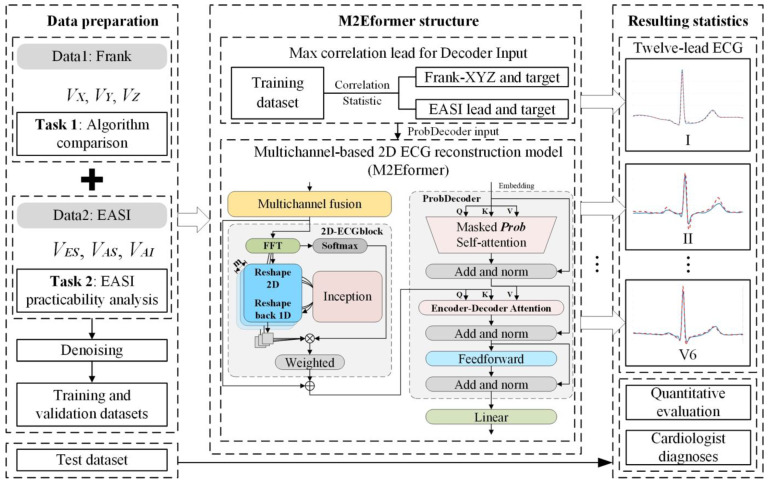
The research framework of this paper. Data1 (Frank-XYZ and 12-lead ECG) was employed to optimize the M2Eformer model and to conduct a comparative performance analysis with prior algorithms. Subsequently, M2Eformer was validated on Data2 to ascertain the reliability of the 12-lead ECG reconstructed from EASI leads for clinical diagnosis.

**Figure 3 bioengineering-11-00293-f003:**
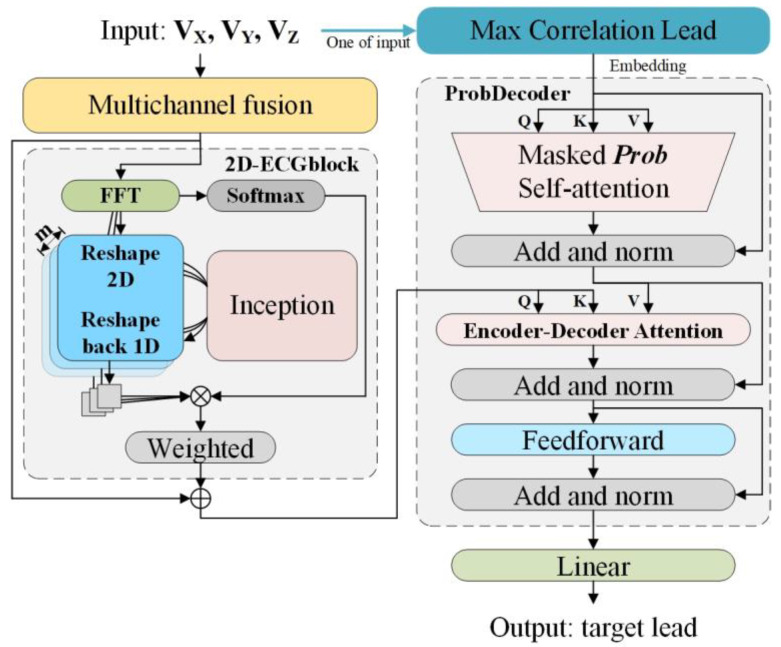
Schematic illustration of the 12-lead ECG reconstruction. Max correlation lead refers to the lead exhibiting the highest correlation with the target lead among the three input leads, as statistically determined based on the training set.

**Figure 4 bioengineering-11-00293-f004:**
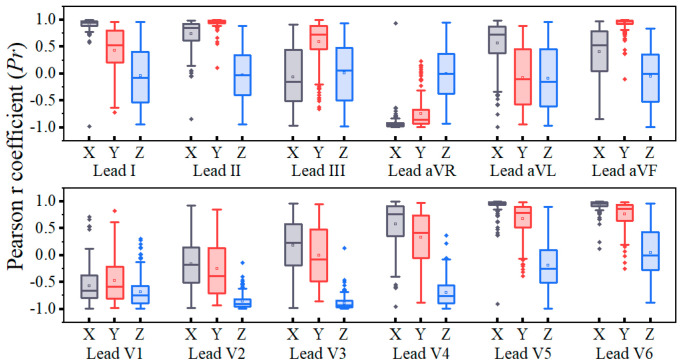
Correlation statistics between the input ECG signal and the target lead.

**Figure 5 bioengineering-11-00293-f005:**
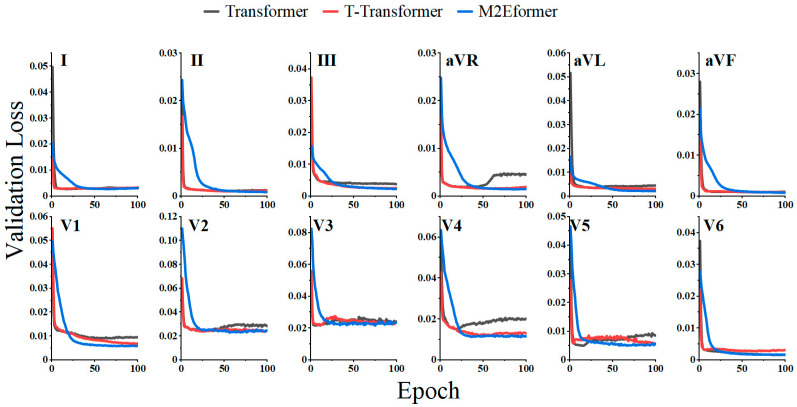
Validation loss of M2Eformer (blue), T-Transformer (red), and Transformer (gray).

**Figure 6 bioengineering-11-00293-f006:**
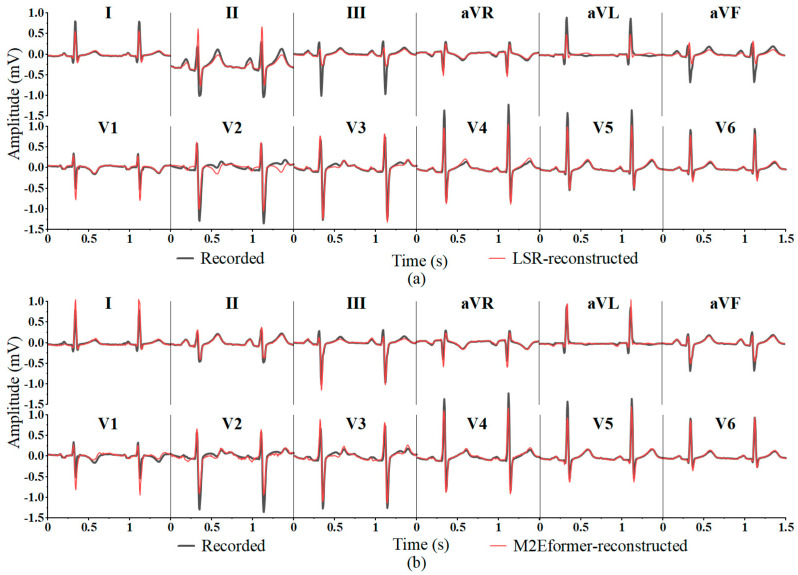
Comparison of the 12-lead ECG reconstructed by (**a**) LSR (**above**) and (**b**) M2Eformer (**below**) in test dataset.

**Figure 7 bioengineering-11-00293-f007:**
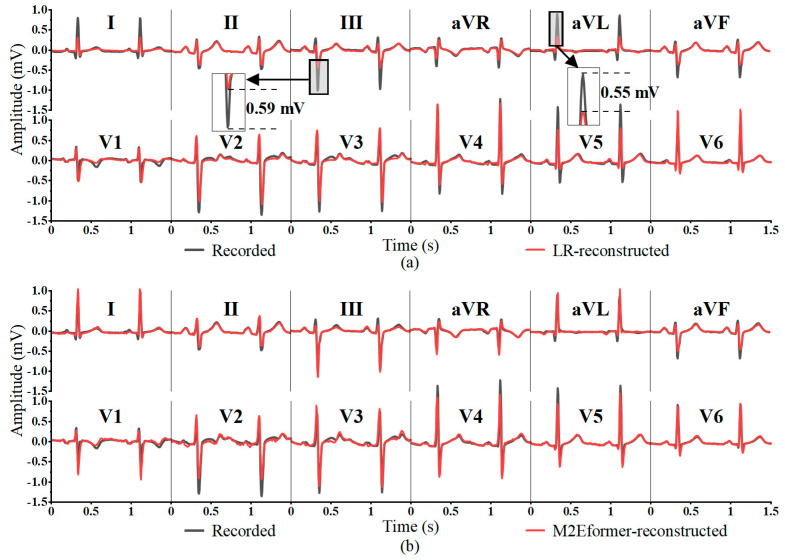
Comparison of 12-lead ECG reconstructed by (**a**) Linear (**above**) and (**b**) M2Eformer (**below**) in the test dataset. (The same segment of the ECG is shown in Figure 6.)

**Figure 8 bioengineering-11-00293-f008:**
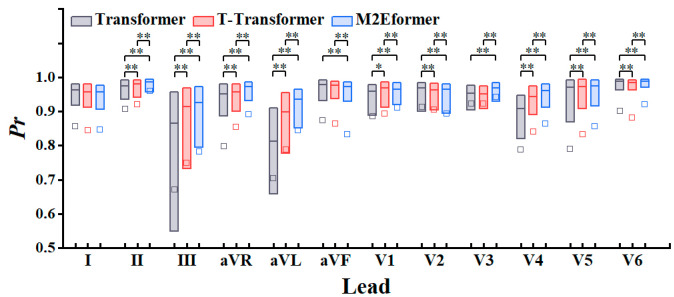
Performance of M2Eformer compared to the Transformer and T-Transformer as correlation coefficient. *: *p* < 0.05, **: *p* < 0.01. The box’s upper edge is the upper quartile; the box’s lower edge is the lower quartile; the box’s middle line is the median value; and the square (□) is the mean value.

**Figure 9 bioengineering-11-00293-f009:**
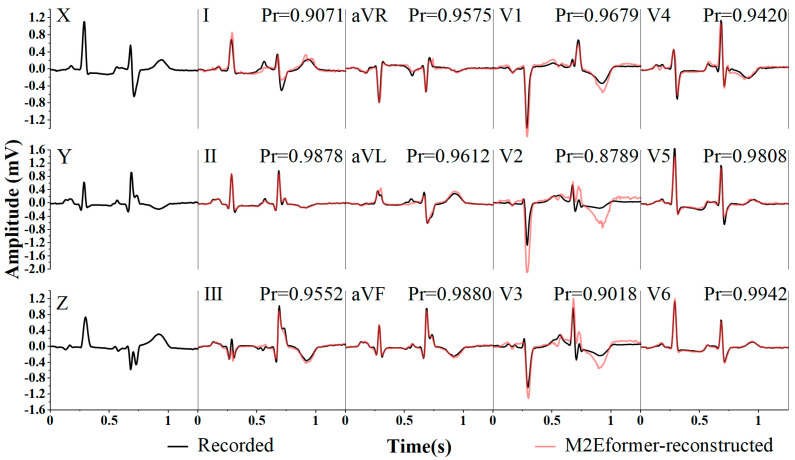
Example of reconstructed 12-lead ECG with Frank-XYZ. Reconstructed (red) and recorded (black) ECG. The test data were collected from MI individuals.

**Figure 10 bioengineering-11-00293-f010:**
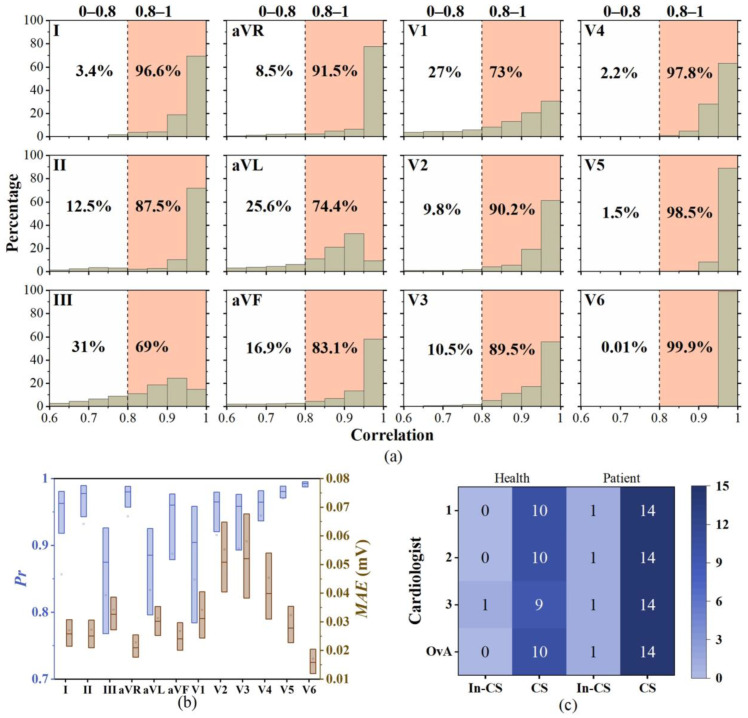
Comparison of the 12-lead reconstructed ECG with EASI vs. the recorded signal. (**a**) The histogram distribution of Pr; (**b**) median (interquartile range) of Pr and MAE; and (**c**) expert labeling results. (In-CS: inconsistent, CS: consistent, OvA: overall agreement). OvA is a method used to analyze the consistency in diagnostic outcomes among three experts as a majority voting mechanism.

**Table 1 bioengineering-11-00293-t001:** Details of the PTB-DN database. The ratio of training to validation to test sets: 3:1:1.

	Training	Validation	Test	Total
Healthy controls	35	11	11	57
Myocardial infarction	127	43	43	213
Bundle branch block	7	2	2	11
Myocardial hypertrophy	6	2	2	10
Valvular heart disease	2	1	1	4
Cardiomyopathy	4	1	1	6
Total	183	61	61	305

**Table 2 bioengineering-11-00293-t002:** Details of the EASI database. The ratio of the training to test sets is 4:1.

	Age ± Std	Training	Test	Total
Healthy controls	26.2 ± 7.2	8	2	10
Atrial flutter	73	12	3	15
Atrial tachycardia	76.5 ± 7.5			
Myocardial infarction	66.2 ± 7.9			
Coronary heart disease	59.3 ± 17.6			
Total	50.5 ± 22.2	20	5	25

**Table 3 bioengineering-11-00293-t003:** Best epoch and minimal validation loss for M2Eformer, T-Transformer, and Transformer.

Model	Best Epoch/Min Loss							
	I	II	III	aVF	V2	V3	V5	V6
M2Eformer	64/0.0027	98/0.0009	97/0.0022	98/0.0008	55/0.0229	74/0.0215	66/0.0050	85/0.0016
T-Transformer	12/0.0028	70/0.0010	91/0.0024	95/0.0009	93/0.0236	10/0.0219	94/0.0054	66/0.0027
Transformer	18/0.0027	51/0.0010	97/0.0038	70/0.0009	28/0.0243	10/0.0215	15/0.0049	89/0.0016

**Table 4 bioengineering-11-00293-t004:** Pearson’s correlation r (*Pr*) in test dataset.

Model	I	II	III	aVR	aVL	aVF	V1	V2	V3	V4	V5	V6	Total
M2Eformer (ours)	0.8465	0.9588	0.7817	0.8921	0.8447	0.8321	0.9105	0.8930	0.9420	0.8641	0.8554	0.9215	0.8785
M2Eformer (ours)	0.8441	0.9207	0.7496	0.8534	0.7875	0.8647	0.8932	0.9052	0.9220	0.8409	0.8326	0.8814	0.8579
Transformer	0.8568	0.9063	0.6705	0.7983	0.7036	0.8748	0.8865	0.9133	0.9217	0.7883	0.7895	0.9004	0.8342
LSTM	0.6573	0.6357	0.4625	0.6181	0.6273	0.6366	0.7009	0.5212	0.5600	0.5341	0.5612	0.6926	0.6006
LSR	0.8507	0.8266	0.5891	0.9224	0.6456	0.6741	0.9015	0.8835	0.9516	0.8552	0.8594	0.9621	0.8268
Linear	0.8087	0.9485	0.6381	0.8642	0.6155	0.8534	0.9068	0.8795	0.9440	0.7759	0.6735	0.9220	0.8192

**Table 5 bioengineering-11-00293-t005:** Mean absolute error (*MAE*) in test dataset.

Model	I	II	III	aVR	aVL	aVF	V1	V2	V3	V4	V5	V6	Total
M2Eformer (ours)	0.0399	0.0215	0.0395	0.0266	0.0370	0.0241	0.0619	0.1044	0.0815	0.0730	0.0629	0.0406	0.0511
M2Eformer (ours)	0.0401	0.0260	0.0368	0.0316	0.0465	0.0252	0.0632	0.0972	0.0879	0.0827	0.0688	0.0465	0.0544
Transformer	0.0380	0.0265	0.0430	0.0343	0.0474	0.0241	0.0638	0.1003	0.0933	0.1045	0.0635	0.0457	0.0570
LSTM	0.0479	0.0576	0.0507	0.0488	0.0373	0.0464	0.0789	0.1410	0.1403	0.1147	0.0853	0.0620	0.0759
LSR	0.0352	0.0422	0.0491	0.0234	0.0401	0.0487	0.0517	0.0980	0.0655	0.0756	0.0570	0.0311	0.0515
Linear	0.0516	0.0212	0.0470	0.0322	0.0478	0.0257	0.0432	0.0753	0.0582	0.0954	0.0997	0.0358	0.0528

**Table 6 bioengineering-11-00293-t006:** Diagnostic results from the cardiologists in the test dataset.

Data	Cardiologist 1		Cardiologist 2		Cardiologist 3		Overall Agreement (OvA)	
	CS/AS	PoC	CS/AS	PoC	CS/AS	PoC	CS/AS	PoC
Healthy controls	9/11	81.8%	9/11	81.8%	10/11	90.9%	9/11	81.8%
Myocardial infarction	41/43	95.3%	41/43	95.3%	39/43	90.7%	43/43	100%
Dysrhythmia	2/2	100%	2/2	100%	2/2	100%	2/2	100%
Bundle branch block	2/2	100%	2/2	100%	2/2	100%	2/2	100%
Myocardial hypertrophy	1/1	100%	1/1	100%	1/1	100%	1/1	100%
Valvular heart disease	1/1	100%	1/1	100%	0/1	0%	1/1	100%
Cardiomyopathy	1/1	100%	1/1	100%	1/1	100%	1/1	100%
Total	57/61	93.4%	57/61	93.4%	55/61	90.2%	59/61	96.7%

AS: all sample; CS: consistent sample; PoC, percentage of consistency.

## Data Availability

The raw data are available on request. For any requests, please contact Songsheng Zhu.

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
