# Peer review of "Wearable 12-Lead ECG Acquisition Using a Novel Deep Learning Approach from Frank or EASI Leads with Clinical Validation"

_bioengineering, 2024, doi:10.3390/bioengineering11030293_

Round 1

Reviewer 1 Report

Comments and Suggestions for Authors

The paper is about Wearable 12-Lead ECG Acquisition Using a Novel Deep 2 Learning Approach from Frank or EASI Leads with Clinical 3 Validation. The paper is nicely written, and I have the following comments:

1 – At the end of Section 1, add a paragraph to summarize the structure of the paper such as: The remaining parts of the paper are structured as: Section 2 present …….

2 – Either add a separate “Related Work” section or add all relevant previous studies in Section 1 and rename this section to Introduction and Related Work

3 – Justify the ration 3:1:1 for the dataset

4 – More information is needed about the deep learning model. What is its specific type? What are the parameters of the model? How are the hyperparameters tuned?

5 – There is no comparison with related studies

6 – In the last section, add the limitations of the proposed model as well as future work

Comments on the Quality of English Language

English is fine

Author Response

Thank you for your comments and insightful suggestions. We have carefully studied your comments and then revised the manuscript accordingly. Our answers are written in red characters and the modifications included in the revised version are given in blue.

Reviewer 2 Report

Comments and Suggestions for Authors

Dear Authors, 

Congrats on the completed well-written and clear manuscript. Here are some considerations to make it better:

- In the introduction, try to quote more, as most information is single-referenced. This helps in getting readership as well. 

-Abstract is good.

- In line 175, elaborate on the criteria used for excluding records.

- Expand Fig 6.

Kudos!

Author Response

(The authors gave the same response as above.)

Reviewer 3 Report

Comments and Suggestions for Authors

An interesting, informative and educational manuscript that has both clinical interest and merit.  The manuscript is well-organized, clearly presents a well-designed study, the acquired data, including figures and tables are reported appropriately.  The conclusions are consistent with the evidence presented in exploring the diagnostic value of using a 12-lead reconstructed ECG through EASI leads for wearable ECG monitoring.  Further, the use of M2Eformer, a novel network architecture for ECG reconstruction, achieved the best overall reconstruction performance for 12 leads.  The references in this article are appropriate.  There are some editing issues in the manuscript that might be considered and address.  The following are suggestions/comments regarding those issues.  Line 100, "... abnormal ECG signal results from the ...".  Line 128, "... of wearable ECG for clinical diagnosis."  Lines 139 & 140, "... 12-lead ECG with standard recorded ones."  Line 193, "cases).  There were also ten healthy participants ...".  Line 204, "... domain energy, thus enabling simultaneous ...".  Line 119, "... evaluations provided by the cardiologist ...".  Line 287, "... of the positions, where the ECG ...".  Line 350, "We also included the commonly ...".  Line 374, "... could potentially lead to a misdiagnosis ...".  Line 385, "... for Linear assessment, which differed significantly.  This result can also result in a misdiagnosis by ...".  Line 386, "... coronary artery disease as myocardial injury)."  Line 411, "... a patient with a myocardial infarction."  Line 540, "... attributed to cardiologist 2's interpreting the ...".   

Author Response

(The authors gave the same response as above.)

Round 2

Reviewer 1 Report

Comments and Suggestions for Authors

The authors addressed my comments

Comments on the Quality of English Language

good